# Mendelian Randomisation Confirms the Role of Y-Chromosome Loss in Alzheimer’s Disease Aetiopathogenesis in Men

**DOI:** 10.3390/ijms24020898

**Published:** 2023-01-04

**Authors:** Pablo García-González, Itziar de Rojas, Sonia Moreno-Grau, Laura Montrreal, Raquel Puerta, Emilio Alarcón-Martín, Inés Quintela, Adela Orellana, Victor Andrade, Pamela V. Martino Adami, Stefanie Heilmann-Heimbach, Pilar Gomez-Garre, María Teresa Periñán, Ignacio Alvarez, Monica Diez-Fairen, Raul Nuñez Llaves, Claudia Olivé Roig, Guillermo Garcia-Ribas, Manuel Menéndez-González, Carmen Martínez, Miquel Aguilar, Mariateresa Buongiorno, Emilio Franco-Macías, Maria Eugenia Saez, Amanda Cano, Maria J. Bullido, Luis Miguel Real, Eloy Rodríguez-Rodríguez, Jose Luís Royo, Victoria Álvarez, Pau Pastor, Gerard Piñol-Ripoll, Pablo Mir, Miguel Calero Lara, Miguel Medina Padilla, Pascual Sánchez-Juan, Angel Carracedo, Sergi Valero, Isabel Hernandez, Lluis Tàrraga, Alfredo Ramirez, Mercé Boada, Agustín Ruiz

**Affiliations:** 1Research Center and Memory Clinic, Ace Alzheimer Center Barcelona, Universitat Internacional de Catalunya, 08017 Barcelona, Spain; 2CIBERNED, Network Center for Biomedical Research in Neurodegenerative Diseases, National Institute of Health Carlos III, 28220 Madrid, Spain; 3Galatea Bio Inc., Hialeah, FL 33010, USA; 4Grupo de Medicina Xenómica, Centro Nacional de Genotipado (CEGEN-PRB3-ISCIII), Universidade de Santiago de Compostela, 15705 Santiago de Compostela, Spain; 5Division of Neurogenetics and Molecular Psychiatry, Department of Psychiatry and Psychotherapy, Medical Faculty, University of Cologne, 50937 Cologne, Germany; 6Department of Neurodegenerative Diseases and Geriatric Psychiatry, University Clinic Bonn, 53127 Bonn, Germany; 7Institute of Human Genetics, School of Medicine & University Hospital Bonn, University of Bonn, 53127 Bonn, Germany; 8Unidad de Trastornos del Movimiento, Servicio de Neurología y Neurofisiología, Instituto de Biomedicina de Sevilla (IBiS), Hospital Universitario Virgen del Rocío/CSIC/Universidad de Sevilla, 41013 Sevilla, Spain; 9Fundació Docència i Recerca MútuaTerrassa, 08221 Terrassa, Spain; 10Memory Disorders Unit, Department of Neurology, Hospital Universitari Mutua de Terrassa, 08221 Terrassa, Spain; 11Hospital Universitario Ramon y Cajal, IRYCIS, 28034 Madrid, Spain; 12Servicio de Neurología, Hospital Universitario Central de Asturias, 33011 Oviedo, Spain; 13Instituto de Investigación Sanitaria del Principado de Asturias (ISPA), 33011 Asturias, Spain; 14Departamento de Medicina, Universidad de Oviedo, 33011 Oviedo, Spain; 15Servicio de Neurología, Hospital Universitario de Cabueñes, 33394 Gijón, Spain; 16Unidad de Demencias, Servicio de Neurología y Neurofisiología, Instituto de Biomedicina de Sevilla (IBiS), Hospital Universitario Virgen del Rocío, CSIC, Universidad de Sevilla, 41013 Sevilla, Spain; 17CAEBI, Centro Andaluz de Estudios Bioinformáticos, 41013 Sevilla, Spain; 18Centro de Biología Molecular Severo Ochoa (UAM-CSIC), 28049 Madrid, Spain; 19Instituto de Investigacion Sanitaria ‘Hospital la Paz’ (IdIPaz), 28029 Madrid, Spain; 20Unidad Clínica de Enfermedades Infecciosas y Microbiología, Hospital Universitario de Valme, 41013 Sevilla, Spain; 21Depatamento de Especialidades Quirúrgicas, Bioquímica e Inmunología, Facultad de Medicina, Universidad de Málaga, 29016 Málaga, Spain; 22Neurology Service, Marqués de Valdecilla University Hospital, University of Cantabria and IDIVAL, 39008 Santander, Spain; 23Laboratorio de Genética, Hospital Universitario Central de Asturias, 33011 Oviedo, Spain; 24Unitat Trastorns Cognitius, Hospital Universitari Santa Maria de Lleida, 25198 Lleida, Spain; 25Institut de Recerca Biomedica de Lleida (IRBLLeida), 25198 Lleida, Spain; 26Departamento de Medicina, Facultad de Medicina, Universidad de Sevilla, 41009 Seville, Spain; 27Instituto de Salud Carlos III, 28222 Majadahonda, Spain; 28CIEN Foundation/Queen Sofia Foundation Alzheimer Center, 28220 Madrid, Spain; 29Fundación Pública Galega de Medicina Xenómica-CIBERER-IDIS, 15705 Santiago de Compostela, Spain; 30German Center for Neurodegenerative Diseases (DZNE), 53127 Bonn, Germany; 31Department of Psychiatry and Glenn Biggs Institute for Alzheimer’s and Neurodegenerative Diseases, San Antonio, TX 78229, USA; 32Excellence Cluster on Cellular Stress Responses in Aging-Associated Diseases (CECAD), University of Cologne, 50931 Cologne, Germany

**Keywords:** Alzheimer’s disease, mosaic loss of chromosome Y, disease progression, GWAS, Mendelian randomization, GR@ACE/DEGESCO, EADB, mild cognitive impairment, polygenic risk score, CSF biomarkers

## Abstract

Mosaic loss of chromosome Y (mLOY) is a common ageing-related somatic event and has been previously associated with Alzheimer’s disease (AD). However, mLOY estimation from genotype microarray data only reflects the mLOY degree of subjects at the moment of DNA sampling. Therefore, mLOY phenotype associations with AD can be severely age-confounded in the context of genome-wide association studies. Here, we applied Mendelian randomisation to construct an age-independent mLOY polygenic risk score (mloy-PRS) using 114 autosomal variants. The mloy-PRS instrument was associated with an 80% increase in mLOY risk per standard deviation unit (*p* = 4.22 × 10^−20^) and was orthogonal with age. We found that a higher genetic risk for mLOY was associated with faster progression to AD in men with mild cognitive impairment (hazard ratio (HR) = 1.23, *p* = 0.01). Importantly, mloy-PRS had no effect on AD conversion or risk in the female group, suggesting that these associations are caused by the inherent loss of the Y chromosome. Additionally, the blood mLOY phenotype in men was associated with increased cerebrospinal fluid levels of total tau and phosphorylated tau181 in subjects with mild cognitive impairment and dementia. Our results strongly suggest that mLOY is involved in AD pathogenesis.

## 1. Introduction

Alzheimer’s disease (AD) is the leading cause of dementia worldwide, accounting for 60–80% of total cases [1]. While Mendelian inheritance is suspected to cause early onset AD (<65 years) [2], late-onset AD (LOAD, >65 years) is a complex, multifactorial disease influenced by both genetic factors and life exposures. The genetic contribution to LOAD is estimated to be 60–80% [3], with *APOE* being the most prominent locus discovered to date [4]. However, demographic features also play a predominant role in AD. Notably, age is considered the most important risk factor for LOAD [5], and women represent nearly two-thirds of the global population with AD [1], showing higher rates of cognitive decline [6,7] than men. However, whether sex should be considered a risk factor for AD or rather a source of disease heterogeneity is a matter of intense debate [8]. Recent reviews have highlighted the importance of reporting results for sex interactions and sex-stratified AD data instead of the more widely used approach of adjusting data by sex [9]. These approaches may help elucidate differences in sex-specific AD risk profiles, which will be of great value in the incoming age of precision medicine.

The male-specific region of chromosome Y is one of the most unexplored regions of the human genome—and it has long been considered a genetic wasteland. Mosaic loss of chromosome Y (mLOY) in blood cells is the most common known form of somatic mosaicism in humans [10,11,12]. Genetic factors together with age, smoking, and other environmental stressors are well-known risk factors for mLOY [13]. Genetic variants associated with mLOY risk are mainly related to mitotic processes, cell cycle regulation, DNA damage sensing and response, and apoptotic processes [14]. mLOY was initially considered a phenotypically innocuous, age-related trait [15,16,17,18]. However, there is increasing evidence that mLOY in blood cells has a direct effect in the aetiopathogenesis of several diseases affecting different tissues. Specifically, blood cell mLOY has been associated with susceptibility to multiple ageing-related diseases, including AD [19], non-haematological cancer [10,20], cardiovascular diseases [21,22], and all-cause mortality risk [10]. The main proposed mechanism to explain blood mLOY pathogenesis is impairment of immune functions caused by the loss of the Y chromosome in leucocytes [23,24,25]. However, it has been described that autosomal genetic predisposition for mLOY is associated with breast cancer in women, indicating that the underlying genomic instability can also explain the associations between mLOY and disease risk [14].

Here, we aimed to study the impact of mLOY on AD risk in the GR@ACE and Dementia Genetics Spanish Consortium (DEGESCO) cohorts [26,27]. First, we checked for blood mLOY associations with AD in a case-control setting and in the phenoconversion process from mild cognitive impairment (MCI) to all-cause dementia and AD. Subsequently, to remove age-confounding effects, we generated an autosomal, age-independent mLOY polygenic risk score (mloy-PRS) and analysed its effect on AD status and progression in both sexes. Finally, we analysed the impact of mLOY in different AD-related biomarkers in cerebrospinal fluid (CSF).

## 2. Results

We calculated the mean log R ratios of probes in the X (LRR-X) and Y (LRR-Y) chromosomes to check the sex chromosome dosages of 7954 clinically reported male samples in the GR@ACE-DEGESCO cohort (Appendix A). We detected one individual with a gain of chromosome Y (GOY, XYY), compatible with a supermale syndrome, and three individuals with Klinefelter syndrome (XXY). We removed women (XX), GOY, Klinefelter individuals, and outliers prior to mLOY computation. For the 7843 remaining XY individuals, we removed second-degree or lower relatives as well as samples with a low genotype call rate (≤0.97) or excess heterozygosity (>3 standard deviations [SD] over the mean heterozygosity of the cohort). We ran principal component analysis to identify the population structure and removed 72 individuals from non-European population (>6 SD from the 1000 Genomes European population mean). We also excluded subjects with detectable autosomal chromosomopathies, (i.e., Down’s syndrome). After applying these exclusion criteria, we split the remaining 6955 male samples into two randomised batches and calculated the mean LRR of probes found at the male-specific region of chromosome Y (mLRR-Y). We used the mLRR-Y_thres_ method of the MADloy R package [28] to call the mLOY status. We did not detect batch effect due to cohort splitting (Appendix A). We excluded 12 additional samples with LRR SD > 0.46. Finally, we plotted mLRR-Y and pseudoautosomal region 1 B-deviation (PAR1-Bdev) values to identify and remove individuals with detectable anomalies in chromosome Y (i.e., partial loss of chromosome Y) or loss of heterozygosity in the PAR1 region (Appendix A). Quality control (QC) and filtering steps for analysis of mLOY phenotypes and mloy-PRS are summarised in Appendix A.

For an initial glimpse at our data, we plotted mLRR-Y values of all AD cases and controls with respect to age (Figure 1). The first thing that became apparent was that our control population is significantly younger than the AD population. Additionally, mLOY occurrence before 65 years was a very rare event in our cohort, indicating that our control population below this age threshold may not be representative for assessing the effect of mLOY on AD. Moreover, our control population mostly lacked individuals older than 85 years. Consequently, we decided to establish a 65–85-year age window to analyse the effect of mLOY on AD. This matches the usual age at onset range for preclinical, prodromal, and mild dementia stages for LOAD in our population [29] and helped reduce the age gap between our case and control groups (Appendix A). Concordant with previous reports, we observed a clear age-related increase in mLOY events in the older individuals (Figure 1). Age was associated with mLOY occurrence in men aged 65–85 years, with an estimated 1% increase in the chance of developing LOY every year (*p* = 3.50 × 10^−11^).

Then, we assessed if continuous mLRR-Y values were differentially distributed among the cases and controls. Both the unadjusted Kolmogorov-Smirnoff test (D = 0.18124; *p* < 2.2 × 10^−16^) and analysis of covariance (ANCOVA) adjusted by age at DNA sampling and *APOE* genotype (F = 68.0, *p* = 2.54 × 10^−16^, Appendix A) yielded highly significant results in the models including all available men in our cohort.

Next, we fitted logistic regressions for AD status using mLOY calls and mLRR-Y, defining three experimental setups: (a) all men with available age at DNA sampling, (b) 65–85-year-old men, and (c) dividing the data in age groups (65–70, 70–75, 75–80, and 80–85 years). We adjusted logistic regressions by age at blood sampling, *APOE* genotype and relevant principal components (PCs) (Appendix A).

We found that the continuous mLRR-Y variable was associated with AD in the group including all men (*N* = 2697, odds ratio (OR) = 2.74, *p* = 0.01), indicating that AD cases had an increased degree of LOY mosaicism compared with controls. We also observed increased mLOY levels in AD men in the 65–85 (*N* = 1944, OR = 2.19, *p* = 0.09) and age-stratified groups with respect to controls, but these differences were not statistically significant (Table 1). Importantly, we noticed that the significant effect observed in the model including all available men could, at least partially, be driven by the dramatic age differences between AD cases and controls in our cohort (Figure 1 and Appendix A) even after adjusting by age. mLOY calls were not significantly associated with AD in the group including all men (*N* = 2697, OR = 1.14, *p* = 0.35), the 65–85-year-old group (*N* = 1944, OR = 1.04, *p* = 0.81), or in the age-stratified groups (Table 1).

To check for an effect of mLOY on risk of conversion to all-cause dementia and AD, we fitted Cox proportional-hazards models adjusted by age at sampling and *APOE* genotype in our prospective cohort of men with MCI (*N* = 400). The continuous mLRR-Y variable had a non-significant risk effect in MCI conversion to all-cause dementia (hazard ratio (HR) = 1.93; *p* = 0.10). The effect size increased when we calculated the model exclusively using conversion to AD but did not reach statistical significance (HR = 2.05, *p* = 0.19). mLOY calls also showed similar but smaller non-significant positive effects for conversion to dementia (HR = 1.17, *p* = 0.40) and AD (HR = 1.38, *p* = 0.20, Figure 2). The Cox model results are summarised in Table 2.

Because the impact of age on mLOY and AD might obscure genuine associations between both phenotypes (mLOY and AD), we decided to construct an mLOY polygenic risk score (mloy-PRS) to evaluate the impact of the genetic variance associated with the mLOY phenotype in AD risk. Our rationale was to implement a Mendelian randomisation strategy reasoning that, if blood cell mLOY is genuinely associated with AD, the genetic factors linked to blood mLOY risk should also be associated with AD and its related endophenotypes. To this end, we generated the mloy-PRS instrument based on a list of autosomal genome-wide significant single-nucleotide polymorphisms (SNPs) associated with the mLOY phenotype identified in a recent genome-wide association study (Appendix A) [14].

For benchmarking purposes of the constructed PRS, we initially validated the effect of mloy-PRS in the mLOY cell phenotype in our cohort (65–85 years old). We fitted a logistic regression for mLOY calls with PRS, age at DNA sampling, and *APOE* genotype as predictors. mloy-PRS (OR = 1.80, *p* = 4.22 × 10^−20^) and age at sampling (OR = 1.08, *p* = 5.07 × 10^−11^) but not *APOE* (OR = 0.88, *p* = 0.15) were significantly associated with mLOY in our population (Appendix A). Importantly, mloy-PRS was orthogonal with age and evenly distributed across the age spectrum (Appendix A). Our results corroborate the validity of mloy-PRS as a Mendelian randomisation instrument for investigating the causal role of mLOY in AD and its endophenotypes and independently confirm the combined risk effect of previously reported loci in the mLOY phenotype [14].

In the case-control setup, we checked the effect of mloy-PRS on AD risk by fitting logistic regressions adjusted by *APOE* genotype, age, and relevant PCs (Appendix A). Interestingly, the effect of mloy-PRS on AD could also be measured in the female samples. Therefore, we established three analysis groups: all (men + women), men only, and women only. We found no association between mloy-PRS and AD in the group including both sexes (Table 3). However, after sex stratification, we found a weak, non-significant, positive effect of mloy-PRS with respect to AD in the male subgroup (*N* = 2471, OR = 1.07, *p* = 0.12), while the effect was mostly neutral in the female subset (*N* = 4 978, OR = 1.00, *p* = 0.93). Next, we assessed the effect of mloy-PRS in disease progression. We adjusted Cox models by age, *APOE* genotype, and cohort ascertainment. We found a male-specific positive effect of mloy-PRS in the disease progression models (N = 682) (Table 3 and Appendix A), with a suggestive signal for MCI-to-dementia progression (HR = 1.11, *p* = 0.08) and a significant risk effect for MCI-to-AD progression (HR = 1.23, *p* = 0.01). Of note, we found no association between mloy-PRS and conversion to all-cause dementia (HR = 0.99, *p* = 0.81) or AD (HR = 0.99, *p* = 0.85) in the female group (*N* = 1082).

Following these results, we proceeded to examine the existence of associations between mLRR-Y and the levels of core AD biomarkers in CSF: Abeta-42, phosphorylated tau 181 (p-tau), and total tau. We only kept individuals aged 65–85 years at the moment of the lumbar puncture (LP), and we excluded those with a gap of >5 years between DNA sampling and the LP (*N* = 214). We adjusted linear regressions by *APOE* genotype, age at LP, and the time window between blood sampling and LP. To account for the effect of syndromic status on the levels of Abeta-42, p-tau, and total tau, we calculated the effect of mLRR-Y in two groups (MCI *N* = 148; dementia *N* = 66) and then performed an inverse-variance weighted fixed-effect meta-analysis. We found that both p-tau (β = 41.92, *p* = 0.01) and total tau (β = 396.69; *p* = 0.004) levels were increased in individuals with a higher degree of mLOY (Figure 3).

Next, we checked for mLRR-Y associations with proteomics data obtained with the Olink ProSeek^®^ multiplex immunoassay for paired plasma and CSF samples in 135 men with MCI. Because mLOY is known to affect the immune system [10,24,30], and inflammation is involved in many processes related to AD pathogenesis [31], we analysed Olink neurology and inflammation panels. We detected inflation in our models (λ = 1.86), with most proteins showing increased levels in the CSF of individuals with a higher degree of blood mLOY (Figure 3). We observed a similar pattern when we analysed the effect of *APOE* genotype and total tau levels, with a large fraction of the proteins showing increased CSF levels in individuals carrying *APOE* risk alleles or displaying higher tau levels, respectively (Figure 3). Moreover, after adjusting our models by total tau, we lost most CSF associations, and the inflation factor was drastically reduced to λ = 0.86 (Figure 3). After covariation with total tau, we found seven nominally significant markers in plasma and one nominally significant marker in CSF. Nevertheless, no proteins passed false-discovery rate (FDR) correction, suggesting that most mLOY associations can be explained by the previously observed correlation between mLOY and tau levels. Summary statistics for association of mLRR-Y to the CSF and plasma proteins are available (Appendix A).

## 3. Discussion

In the present study, we found that MCI men with high genetic risk of developing mLOY have increased chances of progressing to AD over time. The autosomal loci used to construct mloy-PRS had no effect on AD progression in the female subset of our cohort, strongly suggesting that the observed effect is produced via loss of the Y chromosome among men. Importantly, modelling mLOY through its associated genetic variance allowed us to observe mLOY-induced alterations in AD pathogenesis in an age-independent manner, an approach that is unparalleled in previous studies. These results add to previous evidence reporting mLOY as a male-specific AD pathogenic factor.

mLOY is the most common known form of somatic mosaicism among men [10]. Concordantly, we detected mLOY in 18.9% men aged 65–85 years in our cohort. Although classically considered to be a harmless age-related trait, recent studies have revealed that mLOY increases risk of all-cause mortality and several diseases [10,20,21,22]. With such a high prevalence in the older population, interest in determining the effect of mLOY in age-related diseases has increased over the past decade. Previous studies have reported that mLOY is associated to an increased risk and progression rate for AD [19]. A more recent publication claimed that extreme transcriptomic downregulation of chromosome Y decreases AD resilience in men [32]. However, whether mLOY acts as an AD-promoting factor or is just a by-product of ageing needs to be clearly established.

Consistent with previous studies [19], we found a higher degree of mLOY mosaicism (mLRR-Y) in our AD versus control population in unadjusted Kolmogorov-Smirnoff models (D = 0.18124, *p* < 2.2 × 10^−16^) and age-adjusted ANCOVA (F = 68.0, *p* = 2.54 × 10^−16^). We then performed a case-control logistic regression in all available men in our cohort, obtaining significant results (OR = 2.74; *p* = 0.01). Even though we adjusted for age, these results should be interpreted cautiously due to the dramatic age differences between the AD and control groups (Figure 1), as age is an important risk factor for both phenotypes. Thus, aiming to reduce age confounding in our models, we restricted analysis to men aged 65–85 years. However, despite not completely correcting the age gap between the groups, this also reduced our sample size (Appendix A). We found an increased degree of LOY mosaicism (mLRR-Y) in 65–85-year-old cases versus controls (OR = 2.19, *p* = 0.09), but statistical significance was not reached in the models.

Researchers have also found that mLOY increases the rate of AD conversion in MCI men [19]. We selected individuals recruited at the ACE Alzheimer Center Barcelona with an MCI diagnosis at the moment of sampling and available clinical follow-ups and fitted Cox proportional-hazards models. Even though we found risk, i.e., positive, effect directions for mLOY phenotypes towards AD progression, the models were not significant (Figure 2 and Table 2). However, given our small sample size (*N* = 400), we may have lacked sufficient statistical power in this analysis. Remarkably, we noticed that the quantitative mLRR-Y variable performed superiorly in the case-control and disease progression models compared with mLOY calls, implying that if the effects are genuine, the mLOY-induced increase in AD risk and progression may be proportional to the mosaic fraction of LOY cells in blood.

Due to the age-dependent nature of both mLOY and AD, controlling age confounding was very challenging in our cohort. For this reason, we checked mLOY causality in AD by creating an instrument variable and conducted a Mendelian randomisation study. To this end, we generated an age-independent and sex-independent PRS, using 114 independent autosomal genetic variants (Appendix A) previously associated with mLOY [14]. Of note, mloy-PRS successfully predicted mLOY events in our data and was not associated with age or *APOE* genotype (Appendix A and Appendix A). A recently published work found similar effect sizes of this PRS for predicting mLOY calls [33]. Therefore, analysis of mloy-PRS instead of mLOY phenotypes allowed us to overcome the main limitations of the study (age differences and sample size) by providing an age-independent mLOY instrument. This approach allowed us to increase the effective sample size in two ways: (a) by removing the need to restrict analysis to samples with available age at DNA sampling information and (b) by allowing us to introduce all individuals with MCI and subsequent clinical records in disease progression models instead of only those with an MCI diagnosis at the closest clinical evaluation to DNA sampling.

Importantly, we found a male-specific, significant (HR = 1.23; *p* = 0.01) association between mloy-PRS and MCI phenoconversion to AD. Case-control models also reported positive, i.e., risk, effects of mloy-PRS in AD in the male subset (OR = 1.07, *p* = 0.12), but the models did not reach statistical significance even though our sample size was considerably larger in the case-control dataset (*N*_men_ = 2471) than in the longitudinal, prospective MCI dataset (*N*_men_ = 682). These results suggest that mLOY could be more involved in the MCI, early clinical stages of AD aetiopathogenesis than in the preclinical stages of the disease, namely AD risk. However, because the mloy-PRS only explains a fraction of the variance that causes mLOY, a larger sample size may be needed to reach sufficient statistical power to obtain more robust associations in the case-control models. Importantly, mloy-PRS effects were neutral in the groups including women (Table 3), implying that the observed effect of mloy-PRS on AD is unlikely to be driven by the same mechanisms that confer mLOY risk (increased genomic instability and impairment of DNA reparation mechanisms) [14]. Instead, the observed effects are male specific and, therefore, more likely produced via loss of the Y chromosome exclusively in men.

One of the most commonly proposed mechanisms to explain blood cell LOY pathogenesis is the impairment of immune functions [10,14,19]. Interestingly, deregulation of the immune system is one of the hallmark features of AD [34], and genome-wide association studies are revealing an increasing number of genes related to immune functions [35]. LOY has been reported to deregulate the expression of approximately 500 autosomal transcripts in leucocytes [24]. Furthermore, levels of CD99, a cell surface protein involved in several key immune functions, such as leucocyte migration through the vascular endothelium, cell adhesion, and apoptosis [36,37], have been found to be significantly lowered in immune cells with LOY [24,30]. Thus, mLOY-induced alterations in the homeostasis and migration of leucocytes through the brain–blood barrier could explain the observed associations. Additional studies are necessary to corroborate our findings and to identify the potential mechanisms within mLOY that modify AD aetiopathogenesis. Of note, functional restoration of the lost Y-chromosome loci promoting aberrant clonal expansion or transcriptomic deregulation of LOY leucocytes could be an attractive therapeutic strategy to combat AD progression.

One strength of our study is that we modelled LOY through an age-independent PRS instead of just analysing the age-dependent mLOY phenotype and adjusting our data by age. In our opinion, this allowed a clearer and more robust approach for inferring causality between mLOY and AD. We also obtained an independent validation of our findings through AD-related biomarkers, with mLOY phenotypes associated with higher levels of total tau and p-tau in the CSF and displaying the proteomic neurodegenerative biochemical signature observed with other AD-related factors (Figure 3). Higher tau levels are associated with faster rates of cognitive decline [38], supporting the hypothesis that mLOY modulates disease progression. However, our work also faced several limitations: (a) the lack of age at sampling information for most controls and (b) a significantly younger control population compared with the AD population. Both of these limitations ultimately decreased our statistical power to find more robust mLOY–AD associations in the case-control models. We are planning to expand our analysis to additional European population cohorts, which may help us to determine whether mLOY-PRS acts as a male-specific AD risk factor and to confirm the observed effects in disease progression.

We believe Mendelian randomisation analysis is key to confirm causality between mLOY and age-related diseases such as AD, cancer, or cardiovascular disease, where age can also act as a heavy confounder. As we have shown here, sex-stratified Mendelian randomisation can help elucidate whether associations of an mLOY instrument with the outcome are caused by pleiotropy or mediated by the loss of the Y chromosome. Thus, a significant association between an mLOY instrument and a specific outcome in a female sample would likely indicate pleiotropy, as the Y chromosome is absent in women, and the instrument acts as a proxy of genomic instability. For example, in a previous publication, the authors found that their mLOY PRS instrument was associated with breast cancer in women from the UK biobank [14], arguing that these results were reasonable, as genomic instability is a known risk factor for cancer. However, if the association between the mLOY instrument and the outcome is found exclusively in men, or its effect size is significantly greater in men than in women (as both mechanisms could independently increase disease risk), then the Y chromosome loss mediates, at least partly, the observed effect. Encouraging authors to report sex-stratified GWAS summary statistics would open the door to sex-stratified two-sample Mendelian randomisation, which would be a powerful tool to determine the causality of mLOY in the etiopathogenesis of diseases.

In summary, we did not find such strong associations between the blood mLOY phenotype and AD as those reported previously [19]. Due to the demographic features of the GR@ACE-DEGESCO cohort, with older AD patients and younger population-based controls, adjusting our data by age was challenging. Consequently, we modelled the genetic variance associated with mLOY risk, generating a PRS that was associated with MCI conversion to AD in a male-specific manner. This approach allowed us to efficiently control the effect of ageing and to evaluate the potential causality of the mLOY phenotype. Furthermore, lack of association between mloy-PRS and AD in women suggests that the observed effect is produced via the inherent loss of the Y chromosome and that mLOY could be a male-specific AD risk factor. Larger studies may benefit from modelling mLOY using Mendelian randomisation, as case and control populations do not always represent the same age groups in AD cohorts, and the date of DNA sampling of the subjects may not be available.

## 4. Materials and Methods

### 4.1. The GR@ACE-DEGESCO Cohort

The GR@ACE-DEGESCO cohort comprises AD patients and controls from the Spanish population. Patients with AD were collected from the ACE Alzheimer Center Barcelona and 12 other cohorts included in the Dementia Genetics Spanish Consortium (DEGESCO) (Appendix A). Control individuals were provided by the ACE Alzheimer Center (Barcelona, Spain), Valme University Hospital, the Spanish National DNA Bank Carlos III (Salamanca, Spain), and other DEGESCO members. DNA extracted from peripheral blood or saliva (Appendix A) was genotyped in the Spanish National Center for Genotyping (CeGen, Santiago de Compostela, Spain) using the Axiom 815K Spanish Biobank Array (Thermo Fisher), as described previously [26,27].

### 4.2. The ACE MCI-EADB Cohort

The EADB cohort is a prospective cohort comprising individuals with MCI recruited between 2006 and 2013 at ACE Alzheimer Center Barcelona. Briefly, individuals with a clinical dementia rating (CDR) of 0.5 and older than 60 years were selected and underwent at least one follow-up consisting of neurological, neuropsychological, and social work evaluations. A detailed definition of the ascertainment of this cohort has already been described [39,40]. DNA genotyping was performed as described elsewhere [35]. Briefly, DNA extracted from peripheral blood was genotyped with the Illumina Infinium Global Screening Array (GSA, GSAsharedCUSTOM_24+v1.0) at the LIFE & BRAIN CENTER, (EADB node, Bonn, Germany), and SNP genotype calls were obtained from raw probe intensity data in the same centre.

### 4.3. Criteria for AD Diagnosis Case-Control Setup

AD diagnoses were established in all cases by a multidisciplinary working group conformed by neurologists, neuropsychiatrists, and social workers following DSM-IV criteria for dementia and the National Institute on Aging and Alzheimer’s Association’s (NIA-AA) 2011 guidelines for AD definition. In the present study, individuals were labelled as AD when possible or probable AD was endorsed by neurologists at any point of their clinical history. Written informed consent was obtained from all participants. The Ethics and Scientific Committees have approved this research protocol (Acta 25/2016, Ethics Committee. H., Clinic I Provincial, Barcelona, Spain).

### 4.4. Assessment of MCI-To-Dementia/AD Conversion

MCI-to-dementia conversion was determined by integrating the CDR, the global deterioration scale (GDS), and diagnostic assessments at the ACE Alzheimer Center Barcelona, assigned at a consensus conference including neurologists, neuropsychologists, and social workers [29]. Conversion to dementia was defined as the first clinical evaluation reporting a diagnosis of AD [41,42], vascular dementia [43], mixed dementia (AD with cerebrovascular disease), frontotemporal dementia [44,45], or dementia with Lewy bodies [46], combined with a CDR score change from 0.5 to ≥1 and GDS ≥ 4. AD converters were defined as the fraction of converters to dementia that were diagnosed with AD. The baseline criteria varied depending on whether the exposure was mLOY phenotype or its associated PRS. In the first case, baseline was defined as the moment of blood sampling used subsequently for germline DNA extraction, genome-wide genotyping, and mLOY estimation. We selected only those individuals who met Petersen’s criteria [47,48] for amnestic and non-amnestic MCI at the closest clinical evaluation to DNA sampling. Because genotypes used for PRS estimates are invariable, baseline was defined as the patient’s first clinical record meeting Petersen’s criteria for PRS analysis. The follow-up time was defined as the time window between baseline and (a) the date of conversion to dementia (converters) and (b) the date of last clinical evaluation (non-converters). To have a prospective cohort, disease progression models only included individuals who were either originally selected as controls/MCIs in the GR@ACE-DEGESCO case-control cohort or present in the MCI cohort (ACE MCI-EADB).

### 4.5. LOY Determination

We used PennCNV [49,50,51] to process CEL files, following the recommended workflow for Affymetrix arrays [52], to obtain log R ratio (LRR) and B allele frequency (BAF) values for each array probe in our dataset. We determined mLOY by using the MADloy package for R [28]. Briefly, this method estimates mLOY by normalising the mean LRR of probes found at the male-specific region of chromosome Y (mLRR-Y) against the 5% trimmed mean LRR of autosomal chromosomes. Only probes located between PAR1 and PAR2 in chromosome Y, excluding the X transposed region (chrY:6611498-24510581; hg19/GRCh37), are used to compute mLRR-Y. To call mLOY status, we used the mLRR-Y_thres_ method of the MADloy package. Briefly, a threshold is determined by extrapolating the 99% confidence interval of the positive side of the cohort mLRR-Y distribution [10]. Then, samples with mLRR-Y values below the empirically calculated threshold are assigned an mLOY status (or calls). To overcome computational power limitations, we obtained mLOY calls in two randomised batches. We used Bdev, defined as the mean deviation from the expected BAF (0.5) for heterozygous SNPs, in PAR1 (PAR1-Bdev) as a complementary indicator of mLOY (Appendix A).

### 4.6. Sample Processing and QC

We obtained LRR and BAF values for all biallelic markers from 20 068 CEL files (call rate > 0.97 per sample and >0.985 per plate). Then, we retrieved reported male samples and discarded samples with mean LRR-X and LRR-Y corresponding to female (XX) or sex chromosome aneuploidies. Additionally, we removed samples with a high heterozygosity rate, high chromosome X heterozygosity, and population outliers from our dataset. We removed samples with LRR SD > 0.46, a standard QC parameter for Affymetrix LRR data. We used the GENESIS R package [53] to examine relatedness within our dataset. We detected second- or lower-degree relatives by using a kinship threshold of 0.046875 and filtered them out of the dataset. Finally, we removed outliers in the mLRR-Y and Bdev distribution. Specific QC procedures and sample filtering steps for each analysis are summarized in Appendix A.

### 4.7. mloy-PRS

We performed processing, QC, and imputation of the genome-wide SNP data as described elsewhere [27,35]. We calculated mloy-PRS based on independent genome-wide significant variants described previously [14]. Briefly, the authors determined the presence/absence of mLOY in 205011 male samples in the UK biobank and performed a genome-wide association study identifying 18146 variants associated with the mLOY phenotype (*p* < 5 × 10^−8^). Then, they resolved these signals to 156 independent variants by (a) applying LD clumping at 1 Mb and removing correlated signals (*r*^2^ > 0.05) and (b) performing conditional analysis, keeping only secondary signals that reached genome-wide significance before and after conditional analysis. These variants were replicated in 757,114 male samples from European and Japanese ancestry. Out of the 156 reported SNPs, we excluded those unavailable in our dataset, considered rare variants (MAF < 0.01), with low imputation quality (*R*^2^ < 0.3), or located within the sex chromosomes, leaving us with a final number of 114 autosomal SNPs (Appendix A). We calculated mloy-PRS for all individuals in the GR@ACE-DEGESCO and ACE MCI-EADB cohorts by adding the dosage of risk alleles weighted by their reported male-specific effect sizes (beta coefficients). To ease interpretation of results, we standardized mloy-PRS units (SD = 1).

### 4.8. Core AD Biomarkers and Targeted Proteomics

The levels of Abeta-42, tau phosphorylated at position 181 (p-tau), and total tau were measured the same day in CSF samples obtained via LP. The levels were measured using commercially available enzyme-linked immunosorbent assays, namely INNOTEST^®^ β-AMYLOID (1–42), INNOTEST^®^ hTAU, and INNOTEST^®^ PHOSPHO-TAU(181P) (Fujirebio, Spain).

CSF and paired plasma samples collected the same day, as described elsewhere [54,55], underwent targeted proteomics using ProSeek^®^ multiplex immunoassay by Olink Proteomics (Uppsala, Sweden). The protein concentration was measured for 184 proteins included in the commercially available ProSeek^®^ Multiplex panels (inflammation and neurology) in both fluids. QC details and further description of this data are provided elsewhere [56].

### 4.9. Statistical Analysis

We used R software [57] for data processing and analysis. To harmonise effect directions, we multiplied the mLRR-Y variable by −1 due to lower values of mLRR-Y representing a higher degree of mLOY. We fitted logistic regressions adjusted by age, *APOE* genotype, and population structure for case-control analysis. We used the survival R package [58] to fit Cox proportional-hazards models to assess MCI conversion to all-cause dementia or AD. Due to the age-dependent nature of mLOY [11,59], we only included individuals with available age at DNA sampling information for analyses involving mLOY phenotypes. To correct for population structure, we only adjusted by the PCs that were associated with the dependent variable in the models. Thus, we did not adjust for population structure in the Cox models, as PCs showed no effect on disease progression (Appendix A, Appendix A), likely because all MCI samples came from the same centre. We modelled *APOE* genotypes as a continuous variable ranging from −2 to 2, where each *APOE*-ε2 allele contributed with −1, and each *APOE*-ε4 allele added +1, as described previously [40]. To control ascertainment and genotyping bias between MCI cohorts (GR@ACE-DEGESCO & ACE MCI-EADB), we introduced a dichotomous variable in the models testing the association between mloy-PRS and disease progression. For analysis of Olink proteomic data, we adjusted linear regressions by age, the time window between DNA sampling and LP, and *APOE* genotype. Due to the high correlation between the levels of many CSF proteins, total tau, and p-tau (Appendix A), we also included models adjusted by total tau levels. We performed fixed-effect inverse variance weighted meta-analysis with the rma.uni function included in the metafor R package [60].

## Figures and Tables

**Figure 1 ijms-24-00898-f001:**
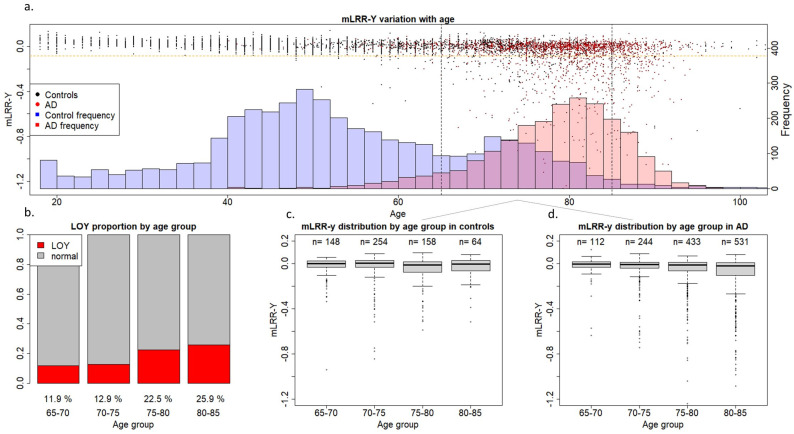
mLRR-y variation with age in the GR@ACE-DEGESCO cohort. (**a**) Age and mLRR-Y distribution in the case and control groups. The dots represent mLRR-Y values for individual samples, and the histogram represents the age distribution across the case and control groups. (**b**) Proportion of individuals with mLOY in the different age groups based on age at blood sampling. (**c**,**d**) mLRR-Y distribution for the different age groups based on age at blood sampling in control and AD individuals, respectively.

**Figure 2 ijms-24-00898-f002:**
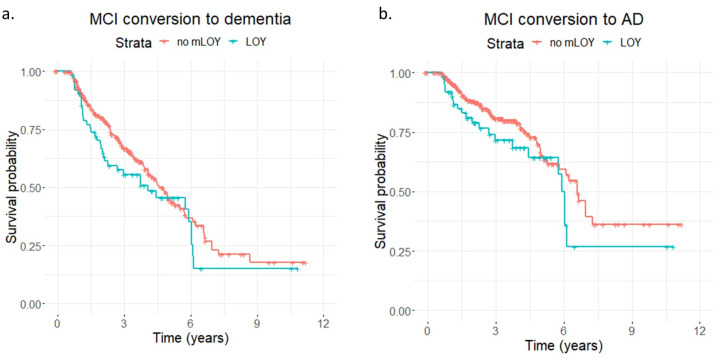
Association of mLOY phenotypes with risk of conversion to dementia and AD dementia over time for men with MCI in the GR@ACE-DEGESCO cohort. Kaplan–Meier plots showing survival time in years for conversion to (**a**) dementia or (**b**) AD for prospective MCI men with LOY (blue) or without LOY (red) in the GR@ACE-DEGESCO cohort.

**Figure 3 ijms-24-00898-f003:**
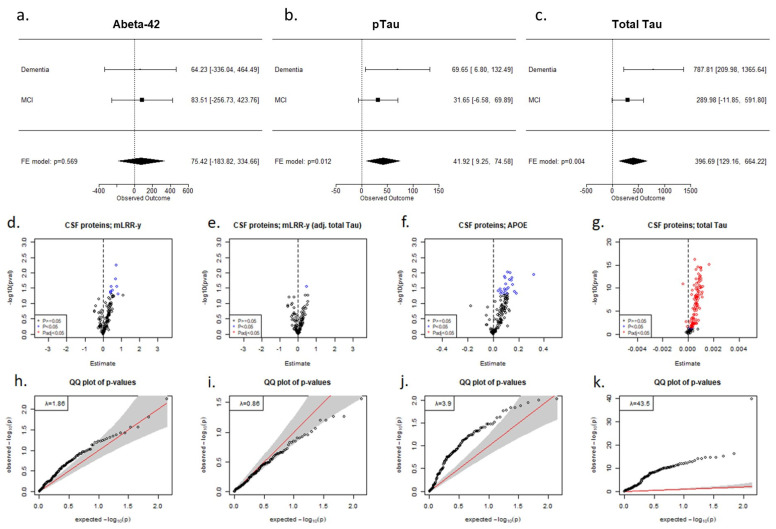
mLRR-Y associations with CSF protein levels. (**a**–**c**) Forest plots showing the effect size obtained in linear regression models for mLRR-Y on (**a**) Abeta-42, (**b**) phospho-tau, and (**c**) total tau in men with MCI or dementia, along with the meta-analysis results. (**d**–**g**) Volcano plots showing association of CSF proteins in the Olink inflammation and neurology panels with (**d**) mLRR-Y, (**e**) mLRR-Y adjusted by total tau, (**f**) *APOE* genotype, and (**g**) total tau. (**h**–**k**) QQ plots obtained in the models for (**h**) mLRR-Y, (**i**) mLRR-Y adjusted by total tau, (**j**) *APOE* genotype, and (**k**) total tau. We adjusted the models by age, the time window between CSF and DNA sampling, and *APOE* genotype.

**Table 1 ijms-24-00898-t001:** Logistic regression results using mLRR-Y and mLOY calls as predictors for AD status. We defined three experimental setups: (a) all men with available age at DNA sampling, (b) 65–85-year-old men, and (c) a stratification of 65–85-year-old men into age groups (65–70, 70–75, 75–80, and 80–85 years old). We adjusted models by age at blood sampling, *APOE* genotype, and PCs. In the age-stratified models, only the effect of mLRR-Y or mLOY are displayed. The 95% confidence interval is presented as the 2.5% quantile (CI2.5) and the 97.5% quantile (CI97.5). OR, odds ratio; SE, standard error.

		All Available Men						65–85-Year-Old Men						Stratified 65–85-Year-Old Men					
		OR	SE	*p*	CI2.5	CI97.5		OR	SE	*p*	CI2.5	CI97.5		Age Group	OR	SE	*p*	CI2.5	CI97.5
**mLRR-Y**	**mLRR-Y**	2.74	0.4	1.13 × 10^−2^	1.29	6.16		2.19	0.47	9.28 × 10^−2^	0.18	1.1		**65–70**	1.47	1.43	0.79	0.07	21.96
	**Age**	1.16	0.01	2.74 × 10^−104^	1.15	1.18		1.27	0.01	6.61 × 10^−70^	1.24	1.31		**70–75**	1.74	0.77	0.48	0.39	8.18
	** *APOE* **	2.52	0.08	1.61 × 10^−32^	2.17	2.94		2.86	0.10	5.40 × 10^−28^	2.38	3.46		**75–80**	2.37	0.77	0.27	0.57	12.16
	**PC1**	2.01 × 10^3^	2.6	3.39 × 10^−3^	12.48	3.29 × 10^5^		0.01	3.33	0.11	0.00	3.48		**80–85**	44.44	2.51	0.13	0.99	2.63 × 10^4^
	**PC2**	0.02	2.58	0.15	0.00	3.8		0.00	3.45	0.12	0.00	4.00		**META**	2.20	0.50	0.12	0.83	5.88
**mLOY calls**	**mLOY**	1.14	0.14	0.35	0.87	1.49		1.04	0.16	0.81	0.76	1.43		**65–70**	0.55	0.45	0.19	0.22	1.30
	**Age**	1.16	0.01	1.11 × 10^−106^	1.15	1.18		1.28	0.01	1.73 × 10^−71^	1.24	1.31		**70–75**	1.17	0.3	0.6	0.65	2.11
	** *APOE* **	2.52	0.08	2.17 × 10^−32^	2.17	2.94		2.86	0.10	6.93 × 10^−28^	2.37	3.46		**75–80**	1.00	0.25	0.99	0.62	1.66
	**PC1**	2.29 × 10^3^	2.59	2.78 × 10^−3^	14.5	3.69 × 10^−5^		0.00	3.32	0.10	0.00	2.96		**80–85**	3.70	0.77	0.09	1.00	24.08
	**PC2**	0.02	2.58	0.14	0.00	3.44		0.00	3.44	0.11	0.00	3.34		**META**	1.03	0.17	0.86	0.74	1.45

**Table 2 ijms-24-00898-t002:** Cox proportional-hazards model results for conversion of men with MCI to all-cause dementia or AD dementia. We adjusted the models by age at sampling and *APOE* genotype. CI, confidence interval.

		Conversion to All-Cause Dementia					Conversion to AD Dementia			
		HR	SE	*p*	95% CI		HR	SE	*p*	95% CI
	Association results for mLRR-Y (continuous mLOY variable)									
**mLRR-Y**		1.93	0.40	0.10	0.87–4.27		2.05	0.55	0.19	0.70–5.97
**Age**		1.10	0.01	3.62 × 10^−15^	1.07–1.12		1.13	0.02	2.38 × 10^−12^	1.09–1.16
**APOE**		1.29	0.13	4.10 × 10^−2^	1.01–1.66		1.56	0.17	8.89 × 10^−3^	1.12–2.17
	Association results for mLOY calls									
**mLOY**		1.17	0.19	0.40	0.81–1.70		1.38	0.25	0.20	0.85–2.24
**Age**		1.10	0.01	1.33 × 10^−15^	1.07–1.13		1.13	0.02	1.62 × 10^−12^	1.09–1.17
**APOE**		1.28	0.13	5.09 × 10^−2^	1.00–1.63		1.54	0.17	1.05E × 10^−2^	1.11–2.13

**Table 3 ijms-24-00898-t003:** Association results for mloy-PRS. (**a**) Results of logistic regressions for case-control AD. We only included individuals aged 65–85 years in the models. We adjusted the models by *APOE* genotype, age, and principal components. (**b**) Results of joint analysis of prospective MCIs in the GR@ACE-DEGESCO and EADB-DEGESCO cohorts using Cox proportional-hazards models for progression from MCI to all-cause dementia or AD. We adjusted the models by *APOE* genotype, age, and cohort ascertainment. The 95% confidence interval is presented as the 2.5% quantile (CI2.5) and the 97.5% quantile (CI97.5). OR, odds ratio; SE, standard error.

(**a**) Logistic regression results:						
	OR	SE	Z	*p*	CI2.5	CI97.5
Case-control, all	1.03	0.03	1.09	0.28	0.98	1.09
Case-control, men	1.07	0.05	1.55	0.12	0.98	1.18
Case-control, women	1.00	0.03	0.10	0.93	0.90	1.10
(**b**) Cox model results:						
	HR	SE	Z	*p*	CI2.5	CI97.5
All, MCI to dementia	1.04	0.04	0.97	0.33	0.96	1.12
All, MCI to AD	1.07	0.05	1.45	0.15	0.98	1.16
Men, MCI to dementia	1.11	0.06	1.77	7.68 × 10^−2^	0.99	1.26
Men, MCI to AD	1.23	0.08	2.53	1.14 × 10^−2^	1.05	1.43
Women, MCI to dementia	0.99	0.05	-0.23	0.81	0.90	1.08
Women, MCI to AD	0.99	0.06	-0.19	0.85	0.89	1.11

## Data Availability

The data presented in this study are available on request from the corresponding author upon reasonable request.

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
