# Peer review of "Mendelian Randomisation Confirms the Role of Y-Chromosome Loss in Alzheimer’s Disease Aetiopathogenesis in Men"

_ijms, 2023, doi:10.3390/ijms24020898_

Round 1
Reviewer 1 Report
The authors present a comprehensive analysis of the role of mosaic loss of chromosome Y (mLOY) in Alzheimer’s disease (AD) etiopathogenesis in males. The authors generated a PRS with genetic variants associated with mLOY risk and found that the PRS was as associated with MCI conversion to AD in a male-specific manner. Here are my concerns/comments:
1. Please list the effect allele frequency of 114 mLOY-associated variants in Supplementary Table 4. Among them, several SNPs in LD with each other, e.g., LD r2=0.04 between rs34890930 and rs17255991, r2= 0.035 between rs74911261 and rs7129527. What are the criteria to select a SNP in constructing the PRS? The effect sizes in this table are male-specific?
2. Although the PRS approach in this study has obvious advantages including the ability to adjust for several covariates in association models, it would be interesting to see the results from a two-sample MR if male AD GWAS summary statistics is available. The sample size for MCI and possible/probable AD may increase for a more powerful estimation. Of course such analysis has limitations too.
Author Response
Dear reviewer,
We greatly appreciate your feedback. Addressing your comments has certainly improved the quality of our manuscript and helped us provide a more complete scope for our work. Please see the attached file to check the modifications made in the text. We have addressed your concerns on the use of English and resubmitted the manuscript to a proofreading service to ensure that English quality is as best as possible. Please find below our answers to your comments:
- Please list the effect allele frequency of 114 mLOY-associated variants in Supplementary Table 4. Among them, several SNPs in LD with each other, e.g., LD r2=0.04 between rs34890930 and rs17255991, r2= 0.035 between rs74911261 and rs7129527. What are the criteria to select a SNP in constructing the PRS? The effect sizes in this table are male-specific?
In table S4 we added the EAF for the UK Biobank males (as reported by the authors of the original publication), as well as the EAFs in the GR@ACE/DEGESCO and EADB-ACE-MCI cohorts. While preparing this information we noticed that, due to an editing mistake, we had reported the reference and alternative alleles, instead of the effect and other alleles corresponding to the betas. We fixed this and now Effect and Other alleles are displayed correctly. In methods [lines 553-560], we added the criteria used by the original authors to determine independence of the 156 reported variants. The effects used for constructing the PRS were male-specific, as we now state [line 565].
- Although the PRS approach in this study has obvious advantages including the ability to adjust for several covariates in association models, it would be interesting to see the results from a two-sample MR if male AD GWAS summary statistics is available. The sample size for MCI and possible/probable AD may increase for a more powerful estimation. Of course such analysis has limitations too.
We agree on the benefits of using two-sample MR approaches. However, as you pointed out, sex-stratified summary statistics are rarely reported, and not available for the most powerful and largest AD GWAS reported to date. Although the increase in sample size may compensate the power reduction caused by having a mix of males and females in the outcome sample, we opted by the PRS approach, using available in-house genetic and clinical data, as it allowed us to perform sex-stratification and test two different hypothesis: mloy-PRS associations in the female group could indicate a pleiotropic effect as genetic variants associated with mLOY are mainly related to genomic instability, while association in males and lack of association in females would be a strong indicator that the effect is mediated by the loss of chromosome Y. We added a paragraph in the discussion [lines 433-449] on this topic, as we consider it enriches our manuscript.
Additionally, please note that we are part of the European Alzheimer’s Disease DNA Biobank (EADB), a recent dataset of 20,464 clinically diagnosed AD cases and 22,244 controls from 15 European countries, and we are collaboratively preparing a manuscript with other researchers using both approaches (two-sample MR and PRS). We also stated this in the new version of the manuscript [lines 429-431].
Kind regards,
Pablo García-González MSc

Reviewer 2 Report
Thank you for your hard work. I can clearly see a lot of work was involved. I only have some minor comments:
1. add some future work
2. can this model be applied to another disease?
Author Response
Dear reviewer,
We greatly appreciate your feedback. Addressing your comments has certainly improved the quality of our manuscript and helped us provide a more complete scope for our work. Please see the attached file to check the modifications made in the text.
- add some future work
As we also pointed out to reviewer #1, we are currently collaborating in the context of the European Alzheimer’s Disease DNA Biobank consortium (EADB), a recent dataset of 20464 clinically diagnosed AD cases and 22244 controls from 15 European countries. We are preparing a manuscript using both the mloy-PRS and a two-sample MR approach to confirm mLOY-AD associations. We added a brief mention to this in the discussion [lines 429-431]. If we confirm mloy-PRS is associated to AD, the next steps will be to investigate the Y chromosome related mechanisms causing AD pathogenicity, as well as for potential protective factors in chromosome Y.
- can this model be applied to another disease?
We have included a paragraph [lines 433-449], where we discuss on how our sex-stratified Mendelian randomization strategy can be very useful to infer causality between mLOY and other age-related disease outcomes. Specifically, because the Y chromosome is only present in males, comparing the association results of an mLOY instrument with the outcome in the male and female groups can help distinguish between pleiotropic effects (genetic variants that increase mLOY risk are related to increased genomic instability) and genuine, Y chromosome loss mediated, effects.
Kind regards,
Pablo García-González MSc
